# Can 3D Imaging Improve Results in Colorectal Cancer Laparoscopic Surgery?

**DOI:** 10.3390/jcm14134437

**Published:** 2025-06-23

**Authors:** Juan Cintas-Catena, Andrea Balla, Javier Valdes-Hernandez, Luis Cristóbal Capitán-Morales, Salvador Morales-Conde, Juan Carlos Gómez-Rosado

**Affiliations:** Department of General and Digestive Surgery, University Hospital Virgen Macarena, University of Sevilla, 41009 Sevilla, Spain; juancintascatena@gmail.com (J.C.-C.); valdeshernandez@hotmail.com (J.V.-H.); lcapitan@us.es (L.C.C.-M.); smoralesc@gmail.com (S.M.-C.); dr.gomez.rosado@gmail.com (J.C.G.-R.)

**Keywords:** laparoscopic surgery, colorectal surgery, three-dimensional (3D) imaging

## Abstract

**Objective:** The aim of this prospective comparative study is to report our experience with 3D laparoscopy in terms of surgeon’s discomfort with 3D vision, and to compare clinical outcomes with 2D laparoscopy in oncological colorectal surgery. **Methods:** From 2016 to 2017, all consecutive patients who underwent elective colorectal surgery for malignancy were enrolled. Based on surgery, patients were grouped as follows: group A, right hemicolectomy; group B, left hemicolectomy; group C, sigmoidectomy; and group D, anterior resection of the rectum. **Results:** In total, 171 patients were included, of which 61 were in group A (45 3D and 16 2D), 18 in group B (15 3D and 3 2D), 44 in group C (30 3D and 14 2D) and 48 in group D (36 3D and 12 2D). The surgeon’s discomfort did not occur due to the 3D vision. Complication rate and mean length of hospital stay (LOS) were lower in the 3D group in comparison to 2D, even if without statistically significant differences, in group B (6.6% versus 66.6% and LOS 6.1 ± 5.2 versus 23 ± 21 days), C (6.7% versus 21.4% and LOS 5.9 ± 2.5 versus 9 ± 8.4 days) and D (27.8% versus 50% and LOS 11.9 ± 16 versus 13 ± 11.8 days), respectively. **Conclusions:** Despite the lack of statistically significant differences between 2D and 3D laparoscopy, this study reports promising trends in favor of 3D laparoscopy, particularly for complex procedures such as anterior resection. Further randomized prospective studies with larger sample sizes and longer follow-up are necessary to conclusively determine the clinical impact of 3D laparoscopy in colorectal surgery.

## 1. Introduction

Colorectal cancer (CRC) treatment is still a debated topic for the surgical community, and currently, it is among the leading health problems [1]. Although the mortality rate has decreased in most developed countries due to medical and surgical advancement, it continues to affect countries with limited resources [1]. The 5-year survival rate is about 65% in the United States and Canada, but it does not exceed 50% in less developed countries [1].

Despite the introduction of the laparoscopic approach for the surgical treatment, which has improved postoperative results in comparison to open surgery in terms of reduced blood loss, less postoperative pain, reduced length of hospital stay (LOS) and quicker return to normal bowel function and daily activities, postoperative complications such as ileus and anastomotic leakage (AL) still represent a challenge for surgeons [2,3,4,5,6]. In particular, AL has been demonstrated to increase hospital stay, cost management and mortality and also affect oncological results [7,8,9,10,11,12].

Although the clinical advantages of laparoscopy in comparison to open surgery are well established [7,8,9], this approach forces surgeons to operate in a three-dimensional space guided by two-dimensional (2D) images [13,14]. As a consequence, surgeons lose depth perception and spatial orientation, and this leads to laparoscopy being more difficult to learn and requiring different psychomotor skills in comparison to open surgery [13,14].

With the aim to reduce the disadvantages of 2D laparoscopic images, three-dimensional (3D) imaging has been developed and introduced in surgery [13,14,15,16]. However, even if 3D introduction in surgery is not a novelty, its introduction and spread in hospitals has not yet been achieved [13,14,15,16]. This could be due to the lack of clear superiority of the 3D systems over 2D in terms of clinical outcomes, but also the observation of side effects when using 3D vision systems such as a degraded viewing condition from poor image resolution, the requirement to wear uncomfortable eyewear and the system’s high cost compared with standard 2D equipment [13,14,15,16].

Nowadays, most of the studies reported in the literature are case series, few of which are about colorectal surgery, and some of them are performed with old 3D technology [13,14,15,16].

The aim of the present prospective study is to report our experience with 3D imaging in laparoscopy in terms of intraoperative adverse events or surgeon’s discomfort for 3D vision, and to compare clinical outcomes between 2D and 3D laparoscopy in oncological colorectal surgery.

## 2. Materials and Methods

This is a comparative prospective observational study. Informed consent from participants and Ethics Committee approval (protocol code 1159-N-16) were obtained.

From October 2016 to November 2017, all consecutive patients who underwent elective colorectal surgery for malignancy by 2D (control group) or 3D (intervention group) imaging laparoscopy in the Unit of Colorectal Surgery, University Hospital Virgen Macarena, Sevilla, Spain, were enrolled in the study. For the control and the intervention group, Karl Storz Image 1S D-Light system (Karl Storz Endoscope GmbH & C. K., Tuttlingen, Germany) and third-generation 3D laparoscopy tower (TC 200 Storz^TM^, Karl Storz Endoscope GmbH & C. K., Tuttlingen, Germany) were used, respectively. The third-generation 3D laparoscopic tower used offers full HD resolution with stereoscopic depth enhancement, improved color contrast, and optimized field of vision for better spatial orientation.

### 2.1. Patients

Patients were included in the study if they were affected by malignant colorectal disease, if they were 18 years old or more, if they underwent elective surgery with curative intention, and if they agreed to sign the informed consent.

Patients were excluded in the case of recurrence and/or preoperative stage IV (according to the American Joint Committee on Cancer—AJCC classification [17]).

Patients were assigned to either 2D or 3D laparoscopy depending on the availability of the system in the operating room. Systematic preference based on patient condition or complexity was not applied. For the aim of the study, patients were divided into 4 groups based on the type of surgery:-Group A: Right hemicolectomy (including enlarged right hemicolectomy for lesion located at hepatic flexure or at proximal transverse colon);-Group B: Left hemicolectomy (including splenic flexure resection);-Group C: Sigmoidectomy;-Group D: Anterior resection of the rectum.

### 2.2. Surgical Technique Details

Surgical procedures were performed by all members of the Unit who perform the same surgical techniques including anastomosis. All surgeons involved had performed more than 100 colorectal laparoscopic procedures using 2D imaging but not in 3D, so the present series coincided with their learning curve with 3D technology.

In the case of laparoscopic right hemicolectomy, an intracorporeal isoperistaltic side-to-side ileo-colic anastomosis is performed with a linear stapler (60 mm purple cartridge), and the enterotomy is closed using a continuous suture with absorbable 2.0 barbed suture (V-loc^TM^, Medtronic, Minneapolis, MN, USA). The specimen is removed through a Pfannesteil incision.

In the case of laparoscopic splenic flexure resection, after division of the inferior mesenteric vein (IMV), the left colic artery (LCA), and the left branch of the middle colic artery (MCA), an intracorporeal isoperistaltic side-to-side colo-colic anastomosis is performed with a linear stapler (60 mm purple cartridge) and the enterotomy is closed using a continuous suture with absorbable 2.0 barbed suture (V-loc^TM^, Medtronic, Minneapolis, MN, USA). The specimen is removed using a Pfannesteil incision.

In the case of laparoscopic left hemicolectomy, IMV and inferior mesenteric artery (IMA) are divided. Using a Pfannesteil incision, left distal colon is removed and divided and a side-to-end anastomosis is performed under laparoscopic vision with a circular stapler (DST series™ EEA™ Stapler, Covidien, Minneapolis, MN, USA).

Finally, in the case of laparoscopic anterior resection, IMV and IMA are divided. Based on the distance of the tumor from the anal verge, partial or total mesorectal excision are performed. The rectal stump is extracted and divided using a Pfannesteil incision. A side-to-end anastomosis is performed with a circular stapler (DST series™ EEA™ Stapler, Covidien, Minneapolis, MN, USA) under laparoscopic vision.

### 2.3. Study Design

Gender, age, body mass index, American Society of Anesthesiologists (ASA) grade, neoadjuvant chemo-radiotherapy (n-CRT), type of surgery, conversions to open surgery, intraoperative complications, intraoperative estimated blood loss (divided in <100 mL, 100–500 mL, and >500 mL), ileostomy creation (protective or ghost ileostomy), operative time, 30-day postoperative surgical complications (graded according to the Clavien–Dindo classification [18]), AL (graded according to Clavien–Dindo classification [18] and to International Study Group of Rectal Cancer, ISGRC classification in the case of anterior resection of the rectum [19]), and LOS were recorded in Microsoft Access program (Microsoft Corporation, Redmond, WA, USA).

AL was defined as any defect in the intestinal wall at the level of the anastomosis leading to communication between the intra- and extraluminal compartments [19].

Postoperative ileus was defined as a temporary decrease in gastrointestinal motility following surgery [20].

Surgeon discomfort was assessed based on self-reported observations immediately after each procedure. Although this method was not based on a validated questionnaire, it provided consistent subjective data across the cohort.

### 2.4. Statistical Analysis

Categorical variables are presented as frequencies and percentages and continuous variables as mean ± standard deviation. Fisher’s exact test and Mann–Whitney U test were employed to evaluate the differences between groups. A *p* value lower than 0.05 was considered statistically significant. Statistical analyses were carried out with SPSS software 25.0 (SPSS Inc., Chicago, IL, USA).

## 3. Results

One hundred and ninety patients underwent elective colorectal surgery for malignant lesion in the study period. Of these, 15 (14 patients who underwent 3D surgery and 1 who underwent 2D surgery) and 4 (1 patient who underwent 3D surgery and 3 who underwent 2D surgery) patients underwent abdominoperineal resection and total colectomy, respectively, and were excluded from the present study, due to the impossibility to obtain a comparison given the small number of patients.

Finally, 171 patients were included, of which 61 patients were in group A (45 3D and 16 2D), 18 in group B (15 3D and 3 2D), 44 in group C (30 3D and 14 2D) and 48 in group D (36 3D and 12 2D) (Figure 1).

In none of the 3D interventions did any adverse events or surgeon discomfort occur due to the 3D vision, and consequently, it was never necessary to convert a 3D intervention to 2D. Statistically significant differences between 3D and 2D in each group did not occur in terms of pre-, intra- and postoperative variables (Table 1).

In patients who underwent the right hemicolectomy with 3D laparoscopy, four conversions to open surgery (8.9%) occurred due to adhesions in two patients, and for parietal and retroperitoneal infiltration in the other two patients. The complication rate for the 3D and 2D groups was 20% and 18.8%, respectively. In the 3D group, nine complications were observed: four postoperative ileus (Clavien–Dindo I), two wound infections (Clavien–Dindo I), two perianastomotic abscess treated by a radiological drainage placement guided by computed tomography (CT) scan (Clavien–Dindo III-a), and one AL treated creating a protective ileostomy (Clavien–Dindo III-b). In the 2D group, three complications were observed: one wound infection (Clavien–Dindo I) and two AL treated surgically with a re-do anastomosis, one of which required intensive care unit hospitalization (Clavien–Dindo III-b and IV-a).

In patients who underwent left hemicolectomy, one patient for each group required conversion to open surgery for parietal infiltration (6.6% and 33.3%, respectively). The complication rate for 3D and 2D groups was 6.6% and 66.6%, respectively. In the 3D group, one AL occurred treated by protective ileostomy creation (Clavien–Dindo III-b), and in the 2D group, two AL occurred both treated by the Hartmann procedure (Clavien–Dindo III-b).

In patients who underwent sigmoidectomy with 3D laparoscopy, one conversion to open surgery (6.7%) occurred due to retroperitoneal infiltration. The complication rate for 3D and 2D groups was 6.7% and 21.4%, respectively. In the 3D group, two complications were observed: one wound infection (Clavien–Dindo I) and one perianastomotic abscess treated by a radiological drainage placement guided by CT scan (Clavien–Dindo III-a). In the 2D group, three complications were observed: one ileus (Clavien–Dindo I) and two AL treated surgically with the Hartmann procedure, one of which died for sepsis (Clavien–Dindo III-b and V).

Among patients who underwent anterior resection of the rectum, six patients in the 3D group and three patients in the 2D group underwent Hartmann procedure without anastomosis. In this group, four (11.1%) and two (16.7%) conversions to open surgery occurred in the 3D and 2D groups, respectively. The reasons for conversion in 3D group were impossibility to tolerate pneumoperitoneum, pelvic bulky mass, infiltration of ureter and uterus infiltration requiring hysterectomy. In the 2D group, conversions occurred due to pelvic bleeding and for an extensive adenopathy at the IMA origin.

Overall, the complication rate for 3D and 2D groups was 27.8% and 50%, respectively. In the 3D group, ten complications were observed: one wound infection (Clavien–Dindo I), one urinary retention (Clavien–Dindo I), one deep surgical site occurrence (SSO) which required a laparoscopic lavage and drainage (Clavien–Dindo III-b), one ileostomy occlusion which required a reshape of it (Clavien–Dindo III-b), one perianastomotic abscess treated by a radiological drainage placement guided by CT scan (Clavien–Dindo III-a, ISGRC classification grade B), and five ALs, four of them treated by Hartmann procedure and one treated laparoscopically with a re-do anastomosis (three Clavien–Dindo III-b, one Clavien–Dindo V as one patient died due to sepsis, five ISGRC classification grade C).

In the 2D group, six complications were observed: one ileus (Clavien–Dindo I), two wound infections (Clavien–Dindo I), one perianastomotic abscess treated by a radiological drainage placement guided by CT scan (Clavien–Dindo III-a), one ileostomy occlusion which required a reshape of it (Clavien–Dindo III-b), and one AL treated by Hartman procedure (Clavien–Dindo III-b, ISGRC classification grade C).

## 4. Discussion

The present study was conducted with the aim of investigating the clinical impact of the use of 3D view during laparoscopic colorectal surgery. Even if in each group no statistically significant differences occurred between 3D and 2D, some interesting findings have to be considered.

First of all, 3D vision did not cause any intraoperative problems to the surgeon in any procedure. Moreover, this study, which included 171 patients across various types of colorectal surgeries, found that even if statistically significant differences in preoperative, intraoperative, or postoperative variables did not occur, the complication rates and mean LOS were lower in the 3D group in comparison to 2D, in group B (6.6% versus 66.6% and LOS 6.1 ± 5.2 versus 23 ± 21 days), C (6.7% versus 21.4% and LOS 5.9 ± 2.5 versus 9 ± 8.4 days) and D (27.8% versus 50% and LOS 11.9 ± 16 versus 13 ± 11.8 days), respectively. Hence, notwithstanding the lack of statistical significance, the introduction of 3D laparoscopy seems to offer benefits in terms of reducing complications and LOS for colorectal surgeries, maybe due to the theoretical advantages of improved depth perception and spatial awareness. Moreover, a closer examination of specific complication types may provide further valuable insights into the subtle benefits or challenges of 3D technology.

The present study was conducted before the introduction of indocyanine green fluorescence technology in our hospital; hence, postoperative results could be affected by the lack of this tool. However, since neither group used this, this is not a bias of comparison.

In the right hemicolectomy group, the complication rates were 20% and 18.8% for the 3D and 2D groups, respectively, with postoperative ileus and AL being the most frequent issues. Although the complication rates were relatively close, the severity of the complications appears to be higher in the 2D group, with more patients requiring re-operation for AL.

In left hemicolectomy and sigmoidectomy, the 3D laparoscopy group showed better overall outcomes, with lower conversion rate to open surgery, postoperative complication rate and fewer severe complications such as AL requiring reoperation.

The most notable advantage of 3D laparoscopy was observed in the group of the anterior resection of the rectum. Here, the complication rate for the 3D group was 27.8%, compared to 50% in the 2D group. More specifically, the 3D group had fewer cases of wound infection, urinary retention, and ileostomy occlusion, and also a lower rate of severe AL. However, this group still faced a considerable number of anastomotic complications, with five cases of AL, one of which led to death from sepsis.

In the present study, in order to avoid bias in the comparison between 2D and 3D, groups have been divided based on type of surgery. Even if this resulted in groups with small numbers of patients, we believe that the comparison is more effective. However, this led us to exclude patients who underwent abdominoperineal resection and total colectomy due to the very small number of patients for each group.

In 2018, the European Association for Endoscopic Surgery (EAES) proposed the first consensus published in the literature about the use of 3D laparoscopic imaging in surgery [21]. Even if the consensus was not specifically related to colorectal surgery, the conclusions reported that 3D image could reduce the operative time in comparison to 2D vision and that further clinical research was required to specifically investigate about the impact of 3D image on complication rate [21]. In the present study, the operative time in 3D groups B, C and D is shorter in comparison to 2D, but with a difference of a few minutes, so data reported in the consensus are confirmed partially [21]. However, as recommended by the consensus, the present study reports detailed differences in complication rate and features, as mentioned above.

Zhan et al., in the largest meta-analysis published until now about the use of 3D imaging in colorectal surgery, including 1160 patients overall (550 patients in the 3D group, and 610 patients in the 2D group) with colorectal malignancy, recruited in 10 articles, report a significant improvement in intraoperative blood loss, operative time, and LOS in the 3D group in comparison to 2D. Moreover, even if without achieving a statistically significant difference, 3D shows better results also in terms of time of pass flatus and number of harvested lymph-nodes in comparison to 2D. Similar results were observed in terms of overall postoperative complication and AL rates between the two groups [22].

Other authors reported about the impact of 3D in specific type of surgery [23,24]. Sapci et al., in a randomized control trial, reported their experience in total colectomy for ulcerative colitis, in 26 and 27 patients who underwent surgery in 2D and 3D, respectively, reporting no statistical differences between groups as in the present study [23]. Portale et al., in a meta-analysis including 275 patients who underwent surgery in 2D and 216 in 3D who underwent right hemicolectomy, did not report any statistically significant differences, even if the operative time was shorter in the 3D group [24].

Finally, although the cost-analysis is not directly addressed in this study, our results and the evidence retrieved from the literature in terms of operative time and LOS could show that 3D laparoscopy has clinical and consequently economic benefits [25].

Nowadays, 3D laparoscopy and its developments have significantly enhanced surgical precision and depth perception [15,22,26,27]. Modern 3D laparoscopic systems now offer improved resolution, reduced latency, and lighter, more ergonomic hardware [15,22,26,27]. Technological advances such as 4K 3D imaging, robotic integration, and augmented reality overlays have further improved the surgeon’s spatial awareness and hand–eye coordination [15,22,26,27]. As a result, 3D imaging is increasingly being adopted in surgical training and in routine clinical practice [15,22,26,27].

The main limitations of the present study are the lack of randomization between 2D and 3D, the lack of standardized assessment of surgeon’s discomfort, the small sample of patients included in each group and the imbalance of patients in some groups. We acknowledge that some subgroups, particularly the left hemicolectomy, were notably unbalanced. This disparity limits the statistical power of subgroup comparisons and should be considered when interpreting the results. Larger and more balanced cohorts are needed to validate subgroup-specific outcomes. However, our study is one of the widest single-center experiences reported in the literature with the use of 3D laparoscopy. Moreover, due to the encouraging results obtained with 3D technology, since its introduction in our center, 2D technology has been abandoned, testifying to the usefulness of this technology and making it impossible to collect further data or comparison.

In conclusion, while the results of this study indicate a statistically significant difference between 2D and 3D laparoscopy in terms of overall complication rates for colorectal cancer surgery, there are some promising trends in favor of 3D laparoscopy, particularly for complex procedures such as anterior resection of the rectum. The improved depth perception and spatial awareness offered by 3D visualization may provide subtle advantages in terms of complication management and precision. Further randomized prospective studies with larger sample sizes and longer follow-up are necessary to conclusively determine the clinical impact of 3D laparoscopy in colorectal surgery.

## Figures and Tables

**Figure 1 jcm-14-04437-f001:**
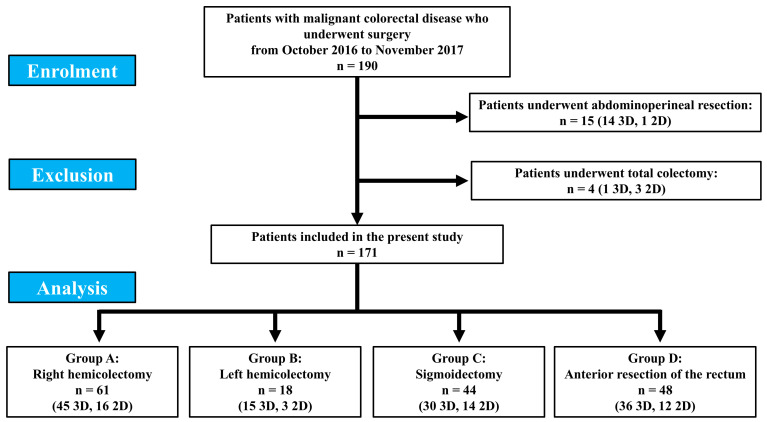
Patient enrolment.

**Table 1 jcm-14-04437-t001:** Results.

	Group A: Right Hemicolectomy			Group B: Left Hemicolectomy			Group C: Sigmoidectomy			Group D: Anterior Resection of the Rectum		
	3D n = 45	2D n = 16	*p* Value	3D n = 15	2D n = 3	*p* Value	3D n = 30	2D n = 14	*p* Value	3D n = 36	2D n = 12	*p* Value
**Sex ratio M:F, n (%)**	27 (60):18 (40)	9 (56.3):7 (43.7)	0.793	9 (60):6 (40)	2 (66.6):1 (33.3)	0.829	20 (66.7):10 (33.3)	10 (71.4):4 (28.6)	0.752	28 (77.8):8 (22.2)	8 (66.7):4 (33.3)	0.441
**Mean age, years ± SD**	69.27 ± 10.9	72.81 ± 8.2	0.282	72.2 ± 13.3	69 ± 4	0.164	69.6 ± 11.3	68.1 ± 13	0.940	66 ± 12	67.3 ± 14.2	0.659
**Mean body mass index, Kg/m^2^ ± SD**	28.58 ± 5.3	28.05 ± 3.2	0.511	28.3 ± 3.9	30.8 ± 6.8	0.738	27.7 ± 4.8	28.3 ± 5.7	0.668	28.3 ± 4.7	27 ± 3.5	0.739
**ASA grade, n (%)**												
I	1 (2.2)	-	1.000	1 (6.6)	-	1.000	-	1 (7.1)	0.318	4 (11.1)	-	0.559
II	12 (26.7)	5 (31.2)	0.752	3 (20)	1 (33.3)	0.426	9 (30)	5 (35.7)	0.738	14 (38.9)	5 (41.7)	1.000
III	31 (68.9)	11 (68.8)	1.000	10 (66.7)	2 (66.6)	1.000	21 (70)	7 (50)	0.313	18 (50)	7 (58.3)	0.743
IV	1 (2.2)	-	1.000	1 (6.6)	-	1.000	-	1 (7.1)	0.318	-	-	1.000
**Neoadjuvant therapy, n (%)**				-	-		-	-	-	23 (46.0)	7 (53.8)	1.000
Chemo-radiotherapy	-	-	-							18 (78.3)	7 (53.8)	0.743
Chemotherapy	-	-	-							-	-	1.000
Radiotherapy	-	-	-							5 (21.7)	-	0.312
**Conversion rate, n (%)**	4 (8.9)	-	0.564	1 (6.6)	1 (33.3)	0.313	1 (6.7)	-	1.000	4 (11.1)	2 (16.7)	0.631
**Intraoperative** **complications, n (%)**	-	-	-	-	-	-	-	-	-	-	-	-
**Estimated blood loss, n (%)**			1.000			1.000			0.170			
>100 mL	34 (75.6)	12 (75)		8 (53.3)	2 (66.6)		19 (63.3)	12 (85.7)		19 (52.8)	5 (41.7)	0.740
100–500 mL	11 (24.4)	4 (25)		7 (46.7)	1 (33.3)		11 (36.7)	2 (14.3)		14 (38.9)	7 (58.3)	0.3193
>500 mL	-	-		-	-		-	-		3 (8.3)	-	0.562
**Protective ileostomy** **creation, n (%)**	-	-	-	-	-	-	-	-	-	7 (23.3) *	5 (55.9) *	0.101
**Mean operative time, minutes ± SD**	146.8 ± 35.7	127.8 ± 33.2	0.110	166.9 ± 33.7	171.7 ± 25.7	0.912	138.8 ± 38.7	142.5 ± 50	0.760	198.2 ± 58	210.8 ± 48.2	0.344
**Complications, n** **(%, Clavien-Dindo** **classification, grade)** Ileus Wound infection Urinary retention Deep Surgical Site Occurrence (SSO) Ileostomy occlusion Abscess Anastomotic leakage	9 (20) 4 (8.9, I) 2 (4.5, I) - - - 2 (4.5, III-a) 1 (2.2, III-b)	3 (18.8) - 1 (6.2, I) - - - - 2 (12.5, III-b)	1.000 0.564 1.000 1.000 1.000 1.000 1.000 0.165	1 (6.6) - - - - - - 1 (6.6, III-b)	2 (66.6) - - - - - - 2 (66.6 III-b)	0.056 1.000 1.000 1.000 1.000 1.000 1.000 0.056	2 (6.7) - 1 (3.3, I) - - - 1 (3.3, III-a) -	3 (21.4) 1 (7.1, I) - - - - - 2 (14.3, III-b, V)	0.306 0.318 0.318 1.000 1.000 1.000 0.318 0.096	10 (27.8) - 1 (2.8, I) 1 (2.8, I) 1 (2.8, III-b) 1 (2.8, III-b) 1 (3.3, III-a) * 5 (16.6, III-b, V) *	6 (50) 1 (8.3, I) 2 (16.7, I) - - 1 (11.1, III-b) 1 (11.1, III-a) * 1 (11.1, III-b) *	0.756 0.250 0.150 1.000 1.000 0.441 0.444 1.000
**Mean hospital stay, days ± SD**	7.5 ± 5.7	7.6 ± 5.6	0.619	6.1 ± 5.2	23 ± 21	0.1	5.9 ± 2.5	9 ± 8.4	0.598	11.9 ± 16	13 ± 11.8	0.259

*: percentages calculated considering that six patients in the 3D group and three patients in the 2D group underwent the Hartmann procedure.

## Data Availability

The original contributions presented in this study are included in the article. Further inquiries can be directed to the corresponding author(s).

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
