# Peer review of "Can 3D Imaging Improve Results in Colorectal Cancer Laparoscopic Surgery?"

_jcm, 2025, doi:10.3390/jcm14134437_

Round 1
Reviewer 1 Report
Comments and Suggestions for Authors
Cintas-Catena et al. conducted a prospective comparative observational study to evaluate the clinical impact of 3D laparoscopy in oncological colorectal surgery, comparing it to 2D laparoscopy. The study assessed surgeon discomfort and various clinical outcomes across different colorectal procedures. While no statistically significant differences were found in overall complication rates or length of hospital stay (LOS), the study reports promising trends favoring 3D laparoscopy, particularly for complex procedures.
Points for Improvement and Clarification:
- Study Design and Patient Allocation to 2D vs. 3D: The study is described as "prospective comparative observational," but states that patients underwent surgery with "2D, or 3D system based on the availability in the operating room".
- Clarification Needed: This implies that the choice of 2D or 3D system was not randomized but rather depended on equipment availability. This non-random allocation can introduce selection bias, as certain patient characteristics or surgical complexities might preferentially be assigned to one system if resources were limited or if surgeons had preferences. Clarifying whether any statistical adjustments were made to account for potential confounding factors arising from this non-random assignment would strengthen the study.
- Surgeon Experience and Learning Curve: While the study notes that "surgeon's discomfort did not occur due to the 3D vision", it doesn't provide specific information on the surgeons' prior experience with 3D laparoscopy.
- Clarification Needed: Given that 3D systems can have a learning curve, detailing the surgeons' baseline experience with both 2D and 3D systems (e.g., number of cases performed before the study) would help contextualize the observed outcomes and ergonomic findings.
- Specifics of 3D Technology: The study mentions using a "third-generation 3D laparoscopy tower (TC 200 Storz)" for 3D imaging.
- Clarification Needed: Providing more technical specifications of the 3D system (e.g., resolution, specific features that enhance depth perception) and discussing how this particular generation of technology might compare to even newer systems would be beneficial for contextualizing the "promising trends" in the rapidly evolving field of surgical technology.
- Sample Size and Imbalance in Subgroups: The authors acknowledge "the small sample of patients included in each group and the imbalance of patients in some groups" as a limitation. For example, in Group B (left hemicolectomy), there were 15 patients in the 3D group but only 3 in the 2D group.
- Clarification Needed: This imbalance can significantly affect the statistical power of subgroup analyses, making it difficult to draw definitive conclusions. Discussing how this imbalance might specifically impact the reliability of the observed trends in smaller subgroups would be important.
- Assessment of Surgeon Discomfort: The abstract states that "Surgeon's discomfort did not occur due to the 3D vision".
- Clarification Needed: It would be helpful to understand how "surgeon's discomfort" was objectively assessed. Was a validated questionnaire used, or was it based on subjective reporting? Providing details on the assessment method would strengthen this finding.
Please cite the following papers:
- 10.1007/s00345-024-05043-9
- 10.1590/S1677-5538.IBJU.2022.0224
Author Response
Cintas-Catena et al. conducted a prospective comparative observational study to evaluate the clinical impact of 3D laparoscopy in oncological colorectal surgery, comparing it to 2D laparoscopy. The study assessed surgeon discomfort and various clinical outcomes across different colorectal procedures. While no statistically significant differences were found in overall complication rates or length of hospital stay (LOS), the study reports promising trends favoring 3D laparoscopy, particularly for complex procedures.
Points for Improvement and Clarification:
- Study Design and Patient Allocation to 2D vs. 3D: The study is described as "prospective comparative observational," but states that patients underwent surgery with "2D, or 3D system based on the availability in the operating room".
- Clarification Needed:This implies that the choice of 2D or 3D system was not randomized but rather depended on equipment availability. This non-random allocation can introduce selection bias, as certain patient characteristics or surgical complexities might preferentially be assigned to one system if resources were limited or if surgeons had preferences. Clarifying whether any statistical adjustments were made to account for potential confounding factors arising from this non-random assignment would strengthen the study.
- Thanks to the reviewer for this observation. The allocation to 2D or 3D laparoscopy was based on availability in the operating room and not randomized. We agree this may introduce a potential selection bias. To minimize this, we ensured that surgeries were assigned independently of patient characteristics, and baseline characteristics between groups were similar, as reported in the Results. Although no formal statistical adjustment was made for allocation, we now explicitly mention this as a potential confounder. We have added this clarification in the “Methods” section and acknowledged this limitation in the “Discussion” section, as follows:
“Patients were assigned to either 2D or 3D laparoscopy depending on the availability of the system in the operating room. Systematic preference based on patient condition or complexity was not applied”.
And
“The main limitations of the present study are the lack of randomization between 2D and 3D, the lack of standardized assessment of surgeon’s discomfort, the small sample of patients included in each group and the imbalance of patients in some groups.”
- Surgeon Experience and Learning Curve: While the study notes that "surgeon's discomfort did not occur due to the 3D vision", it doesn't provide specific information on the surgeons' prior experience with 3D laparoscopy.
- Clarification Needed:Given that 3D systems can have a learning curve, detailing the surgeons' baseline experience with both 2D and 3D systems (e.g., number of cases performed before the study) would help contextualize the observed outcomes and ergonomic findings.
- We agree with the reviewer that surgeon experience is a key factor. All surgeons participating in the study had completed more than 100 colorectal procedures with 2D systems prior to the study. However, the study started the moment the technology arrived in our centre. So, no surgeon had previous experience in 3D surgery until the study started. It has been reported in the Methods, as follows:
“All surgeons involved had performed more than 100 colorectal laparoscopic procedures using 2D imaging but not in 3D, so the present series coincided with our learning curve with 3D technology.”
- Specifics of 3D Technology: The study mentions using a "third-generation 3D laparoscopy tower (TC 200 Storz)" for 3D imaging.
- Clarification Needed:Providing more technical specifications of the 3D system (e.g., resolution, specific features that enhance depth perception) and discussing how this particular generation of technology might compare to even newer systems would be beneficial for contextualizing the "promising trends" in the rapidly evolving field of surgical technology.
- Thank you for pointing this out. We have now added technical details regarding the TC 200 Storz system, including resolution, field of view, and depth enhancement features. However, since the present study is not sponsored, and the aim of our study is not promoting any companies we believe that the information added to the text, as follows, is sufficient:
“The third-generation 3D laparoscopic tower used offers full HD resolution with stere-oscopic depth enhancement, improved color contrast, and optimized field of vision for better spatial orientation.”
- Sample Size and Imbalance in Subgroups: The authors acknowledge "the small sample of patients included in each group and the imbalance of patients in some groups" as a limitation. For example, in Group B (left hemicolectomy), there were 15 patients in the 3D group but only 3 in the 2D group.
- Clarification Needed:This imbalance can significantly affect the statistical power of subgroup analyses, making it difficult to draw definitive conclusions. Discussing how this imbalance might specifically impact the reliability of the observed trends in smaller subgroups would be important.
- We acknowledge this limitation. The imbalance in certain subgroups (e.g., left hemicolectomy) could indeed reduce the power of subgroup analyses. We have included a more detailed discussion about this limitation and emphasized that findings in these subgroups should be interpreted with caution, as follows:
“The main limitations of the present study are the lack of randomization between 2D and 3D, the lack of standardized assessment of surgeon’s discomfort, the small sample of patients included in each group and the imbalance of patients in some groups. We acknowledge that some subgroups, particularly the left hemicolectomy were notably unbalanced. This disparity limits the statistical power of subgroup comparisons and should be considered when interpreting the results. Larger and more balanced cohorts are needed to validate subgroup-specific outcomes.”
- Assessment of Surgeon Discomfort: The abstract states that "Surgeon's discomfort did not occur due to the 3D vision".
- Clarification Needed:It would be helpful to understand how "surgeon's discomfort" was objectively assessed. Was a validated questionnaire used, or was it based on subjective reporting? Providing details on the assessment method would strengthen this finding.
- Surgeon’s discomfort was evaluated through a non-validated subjective report collected after each procedure. We now clarify this in the Methods section and acknowledge the limitation in the Discussion, as follow:
“Surgeon discomfort was assessed based on self-reported observations immediately after each procedure. Although this method was not based on a validated questionnaire, it provided consistent subjective data across the cohort.”
And
“The main limitations of the present study are the lack of randomization between 2D and 3D, the lack of standardized assessment of surgeon’s discomfort, the small sample of patients included in each group and the imbalance of patients in some groups.
- Please cite the following papers:
- 10.1007/s00345-024-05043-9
- 10.1590/S1677-5538.IBJU.2022.0224.
- We regret to not include the suggested articles, but they refer to preoperative 3D reconstruction of image retrieved from CT scan, not to 3D laparoscopic view. Moreover, the two articles referred to nephrectomy not to colorectal surgery and one of the suggested articles is an abstract.

Reviewer 2 Report
Comments and Suggestions for Authors
The article "Can 3D imaging improve results in colorectal cancer laparoscopic surgery?" presents 3D imaging improves the laparoscopic surgery for colorectal cancer in a single centered prospective comparative study. It is well known and well advocated/practiced in clinical setting that 3D imaging is better compared to 2D imaging in regards to colorectal cancer laparoscopic surgery. This article will serve as another addition with new cohort of patients advocating in favor of 3D over 2D imaging.
The outcome of this study although already known in the clinical practice, proves the case in another cohort of patients. The article is well-written, and outcome is well presented and discussed with supporting information. However, few suggestions to improve the manuscript is:
- It is not understandable that why only patients from 2016-2017 are included in the study. Certainly, there seem more room for adding number of patients given that the data is almost decade old. This will provide more statistical power to the study.
- Readers may also be amazed what are recent development and status in this 3D imaging. We can discuss few lines regarding these aspects.
Author Response
The article "Can 3D imaging improve results in colorectal cancer laparoscopic surgery?" presents 3D imaging improves the laparoscopic surgery for colorectal cancer in a single centered prospective comparative study. It is well known and well advocated/practiced in clinical setting that 3D imaging is better compared to 2D imaging in regards to colorectal cancer laparoscopic surgery. This article will serve as another addition with new cohort of patients advocating in favor of 3D over 2D imaging. The outcome of this study although already known in the clinical practice, proves the case in another cohort of patients. The article is well-written, and outcome is well presented and discussed with supporting information.
However, few suggestions to improve the manuscript is:
- It is not understandable that why only patients from 2016-2017 are included in the study. Certainly, there seem more room for adding number of patients given that the data is almost decade old. This will provide more statistical power to the study.
- We appreciate this observation. The data were collected during 2016–2017 as part of an institutional project focused on implementing 3D laparoscopic systems in colorectal cancer surgery. We agree that a larger, more recent dataset would improve statistical power, but in our institution, due to the initial results of this study and the subjective improvements observed, the surgeons transitioned to 3D technology, making impossible to collect data on 2D use nowadays, beyond the period covered by the study. This is reported in the Discussion, as follows:
“Moreover, due to the encouraging results obtained with 3D technology, since its introduction in our center 2D technology was abandoned, testifying to the usefulness of this technology and making it impossible to collect further data or comparison.”
- Readers may also be amazed what are recent development and status in this 3D imaging. We can discuss few lines regarding these aspects.
- A paragraph regarding the 3D laparoscopy status has been added in the Discussion, as follows: “Nowadays, 3D laparoscopy and its developments have significantly enhanced surgical precision and depth perception [15,22,26,27]. Modern 3D laparoscopic systems now offer improved resolution, reduced latency, and lighter, more ergonomic hardware [15,22,26,27]. Technological advances such as 4K 3D imaging, robotic integration, and augmented reality overlays have further improved the surgeon’s spatial awareness and hand-eye coordination [15,22,26,27]. As a result, 3D imaging is increasingly being adopted in surgical training and in routine clinical practice [15,22,26,27].”

Round 2
Reviewer 1 Report
Comments and Suggestions for Authors
No further comments